# Untargeted Metabolomic Analysis and Cytotoxicity of Extracts of the Marine Dinoflagellate *Amphidinium eilatiense* Against Human Cancer Cell Lines

**DOI:** 10.3390/toxins17040150

**Published:** 2025-03-21

**Authors:** María del Carmen Osorio-Ramírez, Alan Gerardo Hernández-Melgar, Allan D. Cembella, Benjamin H. Maskrey, Laura Janeth Díaz-Rubio, Iván Córdova-Guerrero, Johanna Bernáldez-Sarabia, Leticia González-Maya, Baldomero Esquivel-Rodríguez, Celia Bustos-Brito, Alexei F. Licea-Navarro, Lorena M. Durán-Riveroll

**Affiliations:** 1Departamento de Biotecnología Marina, Centro de Investigación Científica y de Educación Superior de Ensenada, Ensenada 22860, Baja California, Mexico; osoriorm@cicese.edu.mx (M.d.C.O.-R.); allan.cembella@awi.de (A.D.C.); 2Departamento de Innovación Biomédica, Centro de Investigación Científica y de Educación Superior de Ensenada, Ensenada 22860, Baja California, Mexico; alangerardo@cicese.edu.mx (A.G.H.-M.); jbernald@cicese.edu.mx (J.B.-S.); alicea@cicese.mx (A.F.L.-N.); 3Department of Ecological Chemistry, Alfred-Wegener-Institut, Helmholtz-Zentrum für Polar-und Meeresforschung, 27570 Bremerhaven, Germany; 4Centre for Environment, Fisheries and Aquaculture Science (CEFAS), Barrack Road, The Nothe, Weymouth DT4 8UB, UK; ben.maskrey@cefas.gov.uk; 5Facultad de Ciencias Químicas e Ingeniería, Universidad Autónoma de Baja California, Tijuana 22424, Baja California, Mexico; ldiaz26@uabc.edu.mx (L.J.D.-R.); icordova@uabc.edu.mx (I.C.-G.); 6Facultad de Farmacia, Universidad Autónoma del Estado de Morelos, Cuernavaca 62209, Morelos, Mexico; letymaya@uaem.mx; 7Instituto de Química, Universidad Nacional Autónoma de México, Ciudad de Mexico 04510, Mexico; besquivel@iquimica.unam.mx (B.E.-R.); celia.bustos@iquimica.unam.mx (C.B.-B.); 8SECIHTI-Departamento de Biotecnología Marina, Centro de Investigación Científica y de Educación Superior de Ensenada, Ensenada 22860, Baja California, Mexico

**Keywords:** anticancer, benthic dinoflagellate, bioactive compounds, cytotoxicity, cell proliferation assay, secondary metabolites, phycotoxins, polyketide

## Abstract

Members of the benthic marine dinoflagellate genus *Amphidinium* produce a variety of bioactive compounds, exhibiting potent cytotoxicity in cell assays. Crude methanolic extracts from three genetically distinct cultured strains of *A. eilatiense* J.J. Lee were screened for cytotoxicity against three human breast and four lung cancer cell lines to evaluate potential applications in anticancer therapy. A standard tetrazolium cell viability assay demonstrated that the methanolic crude extract (100 µg mL^−1^) from strain AeSQ181 reduced cell viability by 20–35% in five cancer cell lines. Further bioassay-guided fractionation of these crude extracts yielded non-polar fractions (FNP-5 and FNP-6) with particularly high cytotoxic activity against lung (H1563) and breast (MDA-MB-231) adenocarcinoma cell lines. Untargeted metabolomic analysis of cytotoxic fractions by liquid chromatography coupled with high-resolution mass spectrometry (LC-HRMS) revealed a much richer chemical diversity profile than previous toxigenicity studies on *Amphidinium* that exclusively focused on linear and cyclic polyethers and their macrolide analogs as putative cytotoxins. This untargeted metabolomic study showed substantial differences in chemical composition between the biologically active and non-active fractions. Preliminary biological and chemical characterization of these *A. eilatiense* fractions confirms that this species is a rich source of bioactive natural products with potential applications such as anticancer therapeutics.

## 1. Introduction

Marine dinoflagellates are a rich reservoir of natural compounds with diverse chemical structures, many showing high bioactivity in cellular and species-ecological interactions [1,2]. Natural products from dinoflagellates have received particular attention over the past few decades because they comprise unusual and unique derivatives of many structural classes, including numerous metabolites with promising biological activity as therapeutics [3]. About 45 classes of natural products are described from dinoflagellates: the principal structural groups are sterols, sphingolipids, carotenoids, cyclic imines, polyenes, polyketide-derived polyethers, and related macrolides [2,4]. Benthic dinoflagellates, in particular those from tropical and sub-tropic coastal waters, yield bioactive secondary metabolites often known as potent toxins [5]. The linear and cyclic polyethers, and their macrolide analogs, have exhibited the highest molecular diversity among benthic dinoflagellates [4,6,7]. These polyketide-derived metabolites have biological activities with complex antagonistic pharmacology, such as antifungal, hemolytic, analgesic, anesthetic, immune-suppressive, neurological, cytotoxic, and anticancer functions [2,8,9,10,11]. Despite the demonstration of cytotoxic activity and evidence of the antiproliferation of cancer cells, research on the bioactive properties of these compounds is still preliminary [8,11,12,13]. Few natural products from dinoflagellates are currently exploited for biotechnological or therapeutic applications [4]. To date, approved pharmacological innovations from dinoflagellate-derived metabolites for the treatment of human diseases remain limited to derivatives of the guanidinium alkaloid saxitoxin for treatment of chronic tension-headache pain and anal fissures [14,15].

The genus *Amphidinium* Claparède et Lachmann is classified as a “naked” (athecate) dinoflagellate, with species distributed from high latitudes to the tropics [16,17]. A few *Amphidinium* species, such as *A. carterae*, *A. operculatum*, *A. massartii*, *A. klebsii*, and *A. gibbosum*, have been confirmed to produce bioactive metabolites, including ichthyotoxins circumstantially implicated in massive fish kills [18,19]. Members of this genus are a uniquely rich source of polyketide-derived amphidinols and structurally related macrolides known as amphidinolides [20,21]. The bioactive properties of these polyketides typically define their mode of action as either ion-channel effectors or enzyme inhibitors affecting metabolic and cell cycle functions.

*Amphidinium eilatiense* J.J. Lee was first isolated and described from the Gulf of Eilat, Israel [22]. In culture, this strain yielded extracts that proved toxic to mice by intraperitoneal injection, with severe signs of respiratory distress and motor activity disturbances. More recently, *A. eilatiense* was also morphologically and genetically characterized from Bahia San Quintín, Baja California, Mexico [23]. Isolated strains in culture have proven to be a rich source of novel and unusual amphidinols [21], some of which have proven cytotoxicity against various cancer cell types [24,25].

Cancer comprises a group of diseases characterized by sustaining proliferative signaling, evading growth suppressors, activating invasion and metastasis, enabling replicative immortality, inducing angiogenesis, and resisting cell death [26]. Compounds from *Amphidinium* species have shown inhibitory effects on colon carcinoma, cervix adenocarcinoma, breast and lung cancer, and lymphocyte B human cell lines. Certain amphidinols [24,25], amphidinolides [27,28], iriomoteolides [29,30,31], and amphirionins [29,30,31,32] inhibit the cell viability of colon carcinoma and cervical cancer cells. In this context, the known cytotoxic, cell viability inhibition, and anticancer activity of *Amphidinium* metabolites are being explored in the prospective development of alternative “small molecule” (<2000 Da) chemotherapeutics for human cancer treatment [33].

The aim of this study was to investigate the cytotoxic effects of *A. eilatiense* extracts and bioassay-guided fractions on seven human adenocarcinoma cell lines, including three breast cancer and four lung cancer lines. The cytotoxic responses were compared to effects on human fibroblast normal cells to assess their potential for therapeutic and pharmacological applications. Three strains of *A. eilatiense* from Bahia San Quintín, Baja California, were selected to evaluate their cytotoxic activity and the metabolomic profile of the most bioactive fractions. The cell lines used in this study were chosen based on previous research [25,34,35] and to represent diverse types of breast and lung cancer, which have some of the highest incidence and mortality rates globally [36,37,38]. This study represents the first effort to link cytotoxicity in cell extracts and fractions from the dinoflagellate *Amphidinium* with components identified by untargeted metabolomic analysis. Liquid chromatography coupled with high-resolution mass spectrometry (LC-HRMS) revealed the high chemical diversity of the fractions exhibiting the highest cytotoxic activity.

## 2. Results

### 2.1. Cytotoxicity Assays of Methanolic Crude Extracts

Methanolic crude extracts of all three strains (AeSQ172, AeSQ177, and AeSQ181) of *A. eilatiense* exhibited cytotoxic activity against all cancer cell lines tested, particularly against human breast and lung carcinoma lines. Equally important, these extracts did not decrease the cell viability of human fibroblast normal cells (MRC-5). Methanolic crude extracts administered at a concentration of 100 µg mL^−1^ induced a statistically significant decrease in cell viability on breast cancer cell lines compared to the negative control (0.1% DMSO). Extract from strain AeSQ177 yielded the most significant decrease in cell viability in human breast carcinoma (T47D) and adenocarcinoma (MDA-MB-231) cell lines, with a 26% and 38% decrease (*p*-value ≤ 0.001), respectively. Strain AeSQ172 and AeSQ177 extracts had a similar effect on the MDA-MB-231 cell line, decreasing cell viability by approximately the same amount. The human breast adenocarcinoma cell line (MCF-7) responded to the treatment with AeSQ181 methanolic extract, by showing a 20% decrease in cell viability (*p*-value ≤ 0.001) (Figure 1a).

When tested against MCF-7 and T47D cell lines, no significant differences were found in the cytotoxic effect of the solvent-free methanolic extracts among the three *A. eilatiense* strains. On the MDA-MB-231 cell line, however, significant differences in effect were observed between AeSQ172 and AeSQ177 in comparison to AeSQ181 extract, with the latter showing a lower cytotoxicity.

Among lung cancer cell lines, the application of solvent-free methanolic crude extracts (100 µg mL^−1^) significantly decreased cell viability for all cancer cell lines compared to the negative control (DMSO 0.1%). The human lung adenocarcinoma cell line (H1563) was most sensitive to all crude methanolic extracts. With this cell line, the AeSQ181 extract was the most potent, decreasing cell viability by 35% (*p*-value ≤ 0.001) (Figure 1b). However, there were no significant differences among effects of AeSQ181 extracts on the other lung cancer cell lines.

None of the extracts caused a significant decrease in viability on the human fibroblast normal cell line MRC-5 when compared to the negative control. Crude extracts from AeSQ177 and AeSQ181 strains induced a statistically significant increase in the cell viability percentage (Figure 1c, indicated with asterisks, *p*-value ≤ 0.001). When all three extracts were compared, however, the *p*-value> 0.05 indicated no significant differences amongst them.

The AeSQ181 methanolic extract was selected for bioassay-guided fractionation due to the following considerations: (1) this strain yielded the highest biomass among the three strains; (2) according to selectivity index results, AeSQ181 strain exhibits higher selectivity against 5 out of 7 tested cancer cell lines (Table 1 and Table 2); (3) among the three dinoflagellate strains, extracts from AeSQ181 demonstrated the lowest negative effect in bioassays on normal cells (MRC-5).

### 2.2. Cytotoxicity Assays of Bioassay-Guided Fractions

Cytotoxicity assays and a metabolomic search for bioactive compounds from strain AeSQ181 were based on methanol-extracted cells. The methanolic crude extract was first partitioned into a polar (aqueous) and a non-polar (ethyl acetate) phase, with a final yield of 1.85 g of the non-polar phase and 1.0 g of the polar phase. Both phases were further fractionated, yielding five polar fractions and 10 non-polar fractions (for reference, see Section 5 (Materials and Methods) and Appendix A).

The most sensitive human cancer cell lines, breast adenocarcinoma (MDA-MB-231) and lung adenocarcinoma (H1563), were selected based on the mean inhibition percentage considering all extracts. The polar fractions had no remarkable cytotoxic activity when tested against cancer cell lines (Appendix A). The non-polar fractions (FNP 1–10) exhibited prominent and statistically significant cytotoxicity against both breast adenocarcinoma MDA-MB-231 and lung cancer cell line H1563 when applied at 50 µg mL^−1^.

Figure 2a shows a decrease in the viability of breast adenocarcinoma MDA-MB-231 cells in response to 8 of 10 tested non-polar fractions. According to the decreasing inhibition potency, the fraction order was F5> F2 > F7 > F6 > F10 > F9 > F1 > F4 > F3 > F8, with F5 yielding a 47% decrease (*p*-value ≤ 0.001) in cell viability (Figure 2a).

Among the ten non-polar fractions tested against the lung adenocarcinoma cell line H1563, only five fractions decreased cell viability in the following decreasing order: F5 > F6 > F2 > F10 > F1, with F5 causing a 55% decrease (*p*-value ≤ 0.001) in cell viability (Figure 2b).

When comparing the cytotoxic effect on cancer and normal cells, the non-polar fractions (FNP-1–5, FNP-7, FNP-8) reduced cell viability by 20% to 40%. In contrast, fractions FNP-6, FNP-9, and FNP-10 did not show any significant effect on cell viability (*p*-value ≤ 0.001) (Figure 2c). Although the statistical analysis indicated significant differences (marked by asterisks) between the groups (*p*-value ≤ 0.001, one-way ANOVA followed by Tukey’s HSD test), there is no significant decrease in cell viability compared to the control group (DMSO 0.1%), represented by the dotted horizontal line.

These results were crucial for bioassay-guided fractionation, as they provided the basis for selecting the most promising fractions for further analysis. Fractions FNP-5 and FNP-6 were chosen due to their higher cytotoxic activity against cancer cells. It is worth noting that FNP-5 reduced cell viability by 16% in the human fibroblast normal cell line, whereas FNP-6 did not affect MRC-5 cell viability (Tukey’s post hoc test) (Figure 2c).

Furthermore, FNP-5 and FNP-6 showed the highest SI values for the MDA-MB-231 breast cancer and H1563 lung cancer cell lines, with some values approaching an index of 2 (Table 2).

### 2.3. Cytotoxicity Assays of Bioactive Sub-Fractions

After the second fractionation of non-polar bioactive *AeSQ181* fractions, three sub-fractions (FNP-5.1, FNP-5.2, and FNP-5.3) were obtained from fraction FNP-5, and two subfractions (FNP-6.1 and FNP-6.2) were derived from FNP-6 (for reference, see Appendix A). The MTS cytotoxicity assays on the MDA-MB-231 cell line indicated that at 50 μg mL^−1^, only sub-fractions FNP-5.1, FNP-5.2, and FNP-6.2 significantly reduced cell viability—by 12% (*p*-value ≤ 0.01), 18% (*p*-value ≤ 0.01) and 13% (*p*-value ≤ 0.001), respectively. Sub-fractions FNP-5.3 and FNP-6.1 demonstrated no cytotoxic effect (Figure 3a). On this breast cancer cell line, sub-fraction FNP-5.2 showed the highest cytotoxic effect; however, no similar cytotoxicity was observed in response to fraction FNP-5.

All five sub-fractions significantly decreased cell viability in the H1563 lung cancer cell line at 50 μg mL^−1^ compared to the negative control (Figure 3b). Among these sub-fractions, FNP-5.1 showed the highest cytotoxic effect, decreasing cell viability by 43% (*p*-value < 0.001) at the same concentration.

### 2.4. Untargeted Metabolomic Analysis of Bioactive Fractions

#### 2.4.1. LC-HRMS Analysis of FNP-5 and FNP-6

An untargeted metabolomic profile of the bioactive non-polar fractions FNP-5 and FNP-6 was obtained from *A. eilatiense* AeSQ181 using LC-HRMS in both positive and negative ionization (ESI +/−) modes. The total ion chromatograms in both ionization modes are shown in Figure 4 and Figure 5, respectively. In the positive ionization mode, 2182 signals were detected in fraction FNP-5 and 3270 in fraction FNP-6; in the negative mode, 1903 signals were identified for FNP-5 and 2730 for FNP-6.

#### 2.4.2. Interpretation of Untargeted Metabolomic Profiles of Bioactive Fractions

Untargeted metabolomic analysis of fractions FNP-5 and FNP-6 from strain AeSQ181 revealed a complex and diverse chemical composition. The chemical landscape analysis utilizing CANOPUS on the NPClassifier revealed six metabolic pathways, comprising 203 features, which were identified in the positive ionization mode from FNP-5. Six metabolic pathways, comprising 203 features, were identified in the positive ionization mode from FNP-5 (Figure 6a). Notably, the fatty acid pathway was the most significant contributor, accounting for 36% of the total. The second most represented pathway was that of alkaloids, comprising 28%, followed by the pathways for amino acids and peptides. Seven metabolic pathways were identified from FNP-6 (Figure 6b), with the primary distinction from FNP-5 being the inclusion of carbohydrates. In FNP-6, as for FNP-5, the fatty acids pathway was the most significant, comprising 22.71% of 1418 features. In second place, amino acids and peptides accounted for 22.64%, followed by polyketides at 19.39%.

In the negative ionization mode, seven metabolic pathways were detected for each fraction, with FNP-5 (753 features) and FNP-6 (1259 features) (Figure 7). Once again, the fatty acids pathway was the most represented with 28.15% in FNP-5 and 30.02% in FNP-6. Alkaloids pathways were the second most represented pathway at 27.49% in FNP-5 and 29.39% in FNP-6. In this ionization mode, amino acids and peptides were the third most represented pathway, with 13.82% in FNP-5. However, in subfraction FNP-6, terpenoids are the third most common.

The molecular network obtained from analyzing fractions FNP-5 and FNP-6 in positive ionization mode (Figure 8a) included 2223 nodes and 1215 connected components. The average number of neighbors, or the average cluster size, is 5.3. Conversely, the network generated for the same fractions in the negative ionization mode (Figure 9a) yielded the following results: 3706 nodes, 2133 connected components, and an average number of neighbors of 4.8. In both ionization modes, the abundance of features was compared to evaluate the similarities and differences between the two active fractions and to assess the effect observed in the biological assay. Fraction FNP-6 displayed more unique features than fraction FNP-5 (Appendix A).

In the chemotaxonomic analysis at the superclass level, thirteen superclasses were identified in both fractions in the positive ionization mode. For FNP-5, the most abundant superclass was lipids and lipid-like molecules with 27.09% (total = 203), whereas for FNP-6, the predominant group was organic acids and derivatives with 27% (total = 1418). The differences between these fractions included the presence of hydrocarbons in FNP-5 and nucleosides, nucleotides, and analogs in FNP-6 (Appendix A).

In contrast, twelve superclasses were identified in FNP-5 and fourteen in FNP-6 in the negative ionization mode. For FNP-5, the most abundant superclass was organic acids (36.92%, total = 753), whereas for FNP-6, lipids and lipid-like molecules represented the most abundant group with 30.96% (total = 1259). A key difference between these fractions was the absence of lignans, neolignans, and related compounds, as well as nucleosides and analogs in FNP-5 (Appendix A).

Untargeted metabolomic analysis revealed that both fractions contained a wide variety of chemical families in both positive and negative ionization mode. A manual search of the databases of features generated by the LC-HRMS analysis for compounds with proven cytotoxic activity, and more specifically, anticancer properties, revealed several relevant targets. In the positive ionization mode (Figure 9), four features previously reported to have cytotoxic activity in various cell lines were identified. One of these features, a self-loop in this molecular network, is erucamide (I), a primary fatty acid amide derived from a fatty acid, which can contain saturated or unsaturated alkyl chains. This compound was detected at a retention time of 23.12 min with a [2M+H]^+^ of the *m*/*z* 675.52, with a mass of 337.26 Da. After annotation by GNPS, it was assigned to the fatty acids’ superclass and “N-acyl amide” class. According to the peak area intensity, Erucamide was present in higher quantities in FNP-6 (20.76%) than in FNP-5 (14.18%).

Ergosta-5,7,22-trien-3-ol (II), a phytosterol annotated by CSIFinger:ID, was also identified but only from one subfraction (FNP-6). This compound, classified within steroids and the Ergostane steroids subclass, yielded a peak area of 18.21%, and was detected at a retention time of 18.07 min with a [M+H-H_2_O]^+^ of *m*/*z* 379.34, for a final molecular mass of 396.34 Da. Additionally, linoleamide (III), a fatty amide with a [M+H]^+^ of the *m*/*z* 280.26 (mass 279.26 Da), was identified only in FNP-6 at a retention time of 17.09 min. Annotated with DEREPLICATOR+, this compound was classified as a fatty amide in the fatty acids’ superclass and N-acyl ethanolamine class. The fatty amide 9-octadecenamide (IV) was identified at 20.29 min with a [M+H-NH_3_]^+^ of *m*/*z* 265.25, with a final mass 281.25 Da. Annotation with GNPS/Spectral Match assigned this fatty amide from the same fatty acid synthesis pathway, as a member of the fatty amide superclass and the N-acyl ethanolamine class. Other fatty amides, such as linoleamide and 9-octadecenamide, were identified in the same cluster (Cluster 2) (Figure 8). All of these fatty amides, which share similar chemical characteristics, were present in relatively high abundance in FNP-6, and, in fact, were unique to this fraction. Notably, some of these features were not annotated by metabolomic tools, but these compounds have reported biological activity, suggesting that they may also share cytotoxic activity effects.

Two clusters and one self-loop were identified in the negative ionization mode analysis (Figure 9). The only self-loop identified was 3,4,5-trimethoxycinnamic acid (V), with an [M-H]^−^ *m*/*z* 237.08 (238.08 Da) and a retention time of 10.13 min. This compound was found in both fractions, with peak areas of 16.2 in FNP-5 and 13.23 in FNP-6. According to the MSFINDER annotation, it is classified as a polyketide, within the cyclic polyketide subclass, and a simple phenolic acids class.

In the first cluster, 1,2-dioctanoyl-sn-glycerol (VI) was identified with an [M-H]^−^ *m*/*z* 343.24, a final mass 344.24 Da, and a retention time of 14.02 min. This compound, annotated by MSFINDER, belongs to the fatty acids’ synthesis pathway, the octadecanoids’ superclass, and the “other” octadecanoids’ class. This fatty acid derivative was found in both subfractions, with similar peak areas of 16.66 in FNP-5 and 18.88 in FNP-6.

In the second cluster, two features were identified: 2′3′-dihydroxyflavone (VII), annotated by GNPS/Spectral Match with an [M-H]^−^ *m*/*z* 255.07 (256.07 Da) and a retention time of 15.19 min; and trihydroxychalcone (VIII), annotated by MSFINDER with an [M-H]^−^ *m*/*z* ratio of 255.07, with a final mass of 256 Da, at a retention time of 15.1 min. Both features were found exclusively in FNP-6, with peak areas of 20.33 and 16.60, respectively.

In summary, 19 features were annotated in the positive mode through the Comprehensive Marine Natural Products Database (CMNPD), which is exclusively marine in origin. Overall, FNP-6 was found to contain a greater number of unique compounds than FNP-5, especially in the positive ionization mode.

## 3. Discussion

### 3.1. Cytotoxicity of Amphidinium Extracts

Many marine microalgae (“phytoplankton”) produce metabolites with cytotoxic effects on various cancer cell lines, making them potential candidates for the development of anticancer therapeutics [39]. Dinoflagellate species in particular are capable of producing a wide range of natural products, including fatty acids, pigments, or polysaccharides, which have demonstrated antiproliferative, cytotoxic, and antitumor activities [8,40]. The cytotoxicity of *Amphidinium* cell extracts has been recognized for several decades based on cell bioassays [25,41], evidence of ichthyotoxicity in laboratory studies [21], and even circumstantial links to fish kills [18,19] in marine ecosystems. In most cases, cytotoxic activity has been associated with large linear polyethers and related macrolides derived from polyketide metabolic pathways, including certain amphidinols (AMs) [24,25], amphidinolides [27,42,43], amphidinins [2], amphirionins [32], colopsinols [44,45], and iriomoteolides [30,31].

Unfortunately, most previous cytotoxicity studies were based only upon crude or semi-purified *Amphidinium* cell fractions, containing a diverse array of poorly chemically characterized polyketides and uncertain quantitation of individual components. The lack of analytical chemical standards, differences in extraction and fractionation methodologies, taxonomic uncertainty regarding species and strain designations, use of diverse cell lines for cytotoxicity, and absence of standardization in quantitation applications pose challenges for valid comparisons of toxicological data among experimental studies. For example, the methanolic extract of *A. carterae* strain CCMP1314 completely inhibited cell viability of human breast adenocarcinoma (MCF-7) and human lung carcinoma (A549) at a concentration of 175 µg mL^−1^ [25]. In comparison, the ethanolic extract of the *A. operculatum* (AA60) culture inhibited the growth of breast adenocarcinoma (MCF-7) and human lung adenocarcinoma (SLKU-1) by 60% at a concentration of 24.7 µg mL^−1^ [34]. The chloroform-extracted fraction from *A. carterae* reduced cell viability in human promyelocytic leukemia (HL60), human melanoma (B16F10), and human lung carcinoma (A549) by approximately 60%, 40%, and 40%, respectively, at a concentration of 50 µg mL^−1^. The same study showed that the ethyl acetate fraction reduced cell viability by approximately 25%, 40%, and 30%, respectively [41]. On the contrary, diethyl ether and aqueous extracts obtained from the benthic dinoflagellate *Ostreopsis* cf. *ovata* (MCCV54) showed cytotoxic activity on the human lung carcinoma (H-460) cell line [46]. When comparing cytotoxic effects, the aqueous extract showed a higher effect, reducing cell viability by 98.8% at a concentration of 1 µg mL^−1^; in comparison, diethyl ether extract only diminished cell viability by 32.5% at the same concentration.

In our current study, the methanolic extracts of *A. eilatiense* AeSQ172 and AeSQ177 exhibited 40% inhibition on human breast adenocarcinoma MDA-MB-231 at a concentration of 100 µg mL^−1^. Among lung cancer cell lines, the cytotoxic effect was consistent across all four types, except for human lung adenocarcinoma (H1563), which exhibited the highest inhibition of cell viability. A comparative study on cytotoxic activity in human promyelocytic leukemia cells (HL-60) [7] demonstrated that the methanolic extract of *A. operculatum* at a concentration of 50 µg mL^−1^ decreased cell viability by approximately 42%.

The above citations illustrate that cytotoxicity results among studies must be interpreted cautiously and can only serve as a guideline for further investigations. The methanolic extracts of all three of the *A. eilatiense* strains showed a lower cytotoxic effect compared to some of the aforementioned studies involving other *Amphidinium* species. The demonstrated level of cytotoxic activity of polar fractions against breast and lung cancer cells are not necessarily consistent with the findings of our study. While this discrepancy highlights the variability in bioactive compound production among different species, it does not diminish the importance or validity of further research on the cytotoxic fractions identified in our work. It does, however, argue for further standardization (toxigenic strains, fractionation schemes, cancer cell-lines, and structural identification of bioactive components) and a major consideration of the cytotoxic mode of action.

The known toxins in *Ostreopsis* cf. *ovata* [47] are mainly ovatoxins (OVTXs) and other structural analogs of palytoxin; one of the most potent non-polymeric toxins. These are also large polyether compounds, but they are not closely structurally related to polyketides such as amphidinols (AM) and macrolides produced by *Amphidinium* [21], which are likely less potent on a molar basis and with a different mode of action. The most relevant comparisons are therefore among variables within an experimental series with defined test strains, fractions, and application concentrations with target cancer cell lines.

Sub-fraction FNP-5.1 showed the highest cytotoxic activity (43% inhibition, see Figure 7). Paradoxically, although high, this response was lower than that induced by the main fraction FNP-5 (55%, see Figure 4) from which FNP-5.1 is derived, resulting in a 12% difference. The observed effect on the MDA-MB-231 adenocarcinoma cell line from the initial fractions was again higher than the effect from the subfractions. When comparing the effect on both cancer cell lines, H1563 displayed a higher inhibition rate than MDA-MB-231 adenocarcinoma. This notable variation in responses could be attributed to synergistic effects from the interaction between different compounds that were only partially separated during the sub-fractioning process. This agrees with the interpretation [35] that observed biological activity from algal extracts could be due to the synergy or the additive potential of several compounds present in the sample.

Despite the differences in the carcinoma cell lines utilized in this study compared to those in previous research, the non-polar fractions derived from *A. eilatiense* exhibited a higher inhibition activity. Some of the apparent inconsistencies in bioactivity between fractions and sub-fractions may be due to the fact that bioactive natural products can have specific therapeutic targets without causing cytotoxic effects [48]. Most importantly, none of the extracts assessed in this current study adversely affected human fibroblast normal cells (MRC-5). This study is unique in that the effects of *Amphidinium* extracts and fractions were tested on both carcinomas and normal cells, which was not addressed in previous studies.

### 3.2. Chemical and Metabolomic Analysis of Bioactive Fractions

Metabolomics is a powerful analytical approach that allows for the characterization and study of the metabolic features, biosynthetic pathways, and underlying mechanisms that shape the cellular interactions of organisms with their environment. Although metabolomics holds great potential for revealing novel natural products from dinoflagellates—toxins, sterols, pigments, and exotic bioactive metabolites—the approach has only been applied in a few cases, e.g., García-Portela et al. (2018) [49], or within an integrated Omics strategy reviewed by Biet et al., (2019) [50], Beedessee et al., (2020) [51], and Roussel et al. (2023) [52]. Advanced mass spectrometry techniques, such as combinations of liquid chromatography-coupled mass spectrometry (LC-MS), tandem mass spectrometry (LC-MS-MS), and high-resolution mass spectrometry (LC-HRMS), are indispensable in the analysis of such chemically complex mixtures for metabolomics. These methods are particularly effective for separating, identifying, quantifying, and analysis of metabolites within semi-purified mixtures [53].

Non-metabolomic studies regarding cytotoxicity comparisons among *Amphidinium* species and strains have focused on the putative bioactivity of amphidinols (AM), the best known and characterized group of bioactive metabolites from this genus. In fact, strain selection (AeSQ172, AeSQ177, and AeSQ181) for the current experiments on cytotoxic activity against breast and lung cancer cell lines was based primarily on the high diversity and cell quota of AMs produced among these strains [21], some of which exhibit cytotoxic activity [24]. For example, all three strains produce the proven cytotoxic compound [24] amphidinol 02 (AM02). Significantly, the present metabolomic study did not reveal any polyketide members of the amphidinol chemical family in the bioassay-guided fractions. We conclude that AMs are not exclusively responsible for the reported cytotoxicity among *Amphidinium* strains, but the bioactive congeners remain elusive and unidentified.

Metabolomic analysis of bioactive fractions of *A. eilatiense* AeSQ181 revealed a wide range of chemical features with potential, not only for cytotoxicity, but also for other biological activities. Comparing fractions FNP-5 and FNP-6, it is clear that the latter is chemically more complex, and results from both ionization modes indicate that strain AeSQ181 employs multiple biosynthetic pathways to produce natural products with biological potential. A total of 19 features were annotated in the positive mode through the Comprehensive Marine Natural Products Database (CMNPD). For example, two identified features (Noyapyrone A and Melearoride A) from AeSQ181 have been reported to show antibacterial and antifungal activity, respectively.

Intriguingly, after the metabolomic analyses, no amphidinols were found in FNP-5 or FNP-6, despite the fact that the strains chosen for this study are recognized for their significant production of these bioactive polyketides [21]. Nevertheless, there is no doubt about the effect of these strain extracts, fractions, and subfractions on cancer cell line viability. One probable explanation for this apparent paradox is that supplementary or synergistic cytotoxicity are attributable to a different chemical group. Based on the metabolomic analyses of both bioactive fractions, we know that the fatty acids pathway yields the most abundant components identified in both ionization modes. Various types of fatty acids, such as n-3 polyunsaturated fatty acids or PUFAs, eicosapentaenoic acid (EPA), and docosahexaenoic acid (DHA), are known to disrupt cell membranes [54,55], inducing apoptosis in cancer cells. This disruption occurs due to their capability to integrate into the lipid bilayer, altering the structural cohesion of the membrane and enhancing anti-neoplastic effectiveness [56,57]. Therefore, the cancer cells’ response to the subfractions, even in the absence of known amphidinols or other polyketides with proven anticancer activity, could arise from these fatty acids, influencing the cell membranes in ways that warrant further research.

Molecular networking is based upon paired spectral alignment with a similarity algorithm for spectral cosine. As applied in our metabolomic analysis, modified spectral similarities are searched for to compare fragmentation spectra (MS2) of ions with identical *m*/*z* values, as well as MS2 spectra compensated by the same *m*/*z* subtraction result as the precursor ion [58,59]. The chemical features highlighted in this study were selected based on their previously reported cytotoxic activity, some of which exhibit anticancer properties against various tumor cell lines. For example, 9-octadecenamide (IV), located in Cluster 2 in the positive mode (Figure 8), derived from oleic acid, induces cell mortality in hepatocellular carcinoma cells (HepG2) [60]. Another relevant compound within the same cluster (Figure 8), linoleamide (III), exhibits cytotoxic activity as an antagonist of the TRPV2 calcium membrane channel. This TRPV2 channel, expressed in various cancer cell lines, plays a crucial role in survival and metastasis signaling pathways. In contrast, Erucamide (I), another fatty amide found in a self-loop, shares the same TRPV2 antagonist activity, but it is located in a different region of the molecular network. Erucamide (I) and Cluster 2, where linoleamide (III) is located, are quite far apart in the chemical space represented by the molecular network, but these amide have similar chemical and biological properties [61]. This apparent chemical redundancy could complicate the process of detection and isolation of compounds with biological activities of interest. Redundancy reflects the high complexity within the chemical space for natural products produced by these dinoflagellate strains, rendering a promising path ahead for further inquiries and discoveries on bioactive signatures.

As dinoflagellates comprise a rich source of novel and unusual sterols, ergosta-5,7,22-trien-3-ol (II), more commonly known as ergosterol or Provitamin D2, is highlighted from Cluster 1. This compound elicits cytotoxic activity and can arrest the cell cycle in myeloma cells [62].

Finally, regarding key metabolic compounds detected in negative ionization mode, the generated molecular network yielded two clusters and a self-loop. The compound 1,2-dioctanoyl-sn-glycerol (VI) (Figure 9 has been evaluated in an affinity assay for the TRPC7 membrane protein [60]. This protein is implicated in various forms of metastatic cancer, including breast, prostate, pancreatic cancer, as well as leukemia, and neuroblastoma [63], making it a top priority for further research and screening.

## 4. Conclusions

This study is a significant contribution to understanding the chemical complexity hidden in the metabolism of *Amphidinium eilatiense* and dinoflagellates in general. The challenge remains to further link the diversity of its metabolites more closely to the biological activity observed. Through bioassay-guided fractionation, we demonstrated that crude methanolic extracts, as well as their solvent-free fractions and subfractions, possess cytotoxic activity against breast and lung cancer cell lines. Although these cell lines belong to the same disease category, their differentiated molecular profiles outline the importance of considering biological variability in bioactivity studies. Metabolomic analysis of chemical diversity indicates that bioprospecting for bioactive natural products in *Amphidinium* has been too limited, with the focus only on amphidinols (AM) and chemically related polyketides known to be toxic in biological systems. Despite the observed cytotoxic effects, no adverse responses were detected on healthy cells, which highlights the selective therapeutic potential of these compounds.

This study presents a classification, and an exhaustive identification, of the secondary metabolites found in the non-polar fractions with the highest cytotoxic activity. As such, this demonstrates the utility of advanced platforms like Global Natural Products Social Molecular Networking (GNPS) and other computational (in silico) tools.

A future challenge derived from the results of this study is accomplishing the isolation of the annotated compounds and testing them against the most sensitive cell line. This would allow them to find their molecular targets, and so, unveil the action mechanism of these compounds with pharmacological potential. The presence of unidentified features or metabolites non-grouped in nodes represents a clear opportunity for future studies that can explore the chemical diversity of *A. eilatiense* in-depth and its potential as a source of new natural products with cytotoxic activity. In this sense, the future perspectives aim at a deepening of the exploration of the metabolome, where the use of advanced computational tools will facilitate the annotation of new compounds.

## 5. Materials and Methods

### 5.1. Dinoflagellate Culture and Biomass Harvest

Monoclonal strains of *A. eilatiense* (AeSQ172, AeSQ177, and AeSQ181) for these experiments were obtained from Bahía de San Quintín, Baja California, Mexico (30°27′14″ N, 116°00′15″ W) in December 2019. Details of field collection, cell isolation, and culturing techniques were as described in [23]. The cells were cultivated in GSe medium [64] modified without soil extract and prepared with autoclaved (121 °C, 15 min) 1 μm-filtered seawater at salinity 36. Monoclonal cell cultures were cultured in 1 L Erlenmeyer polycarbonate flasks and, according to cell density, transferred from an initial volume of 50 mL to a final volume of 700 mL. The cultures were maintained between 17 and 28 days at 23 ± 2 °C under a 12:12 light:dark cycle and illumination of 50 µmol photons m^−2^ s^−1^.

Cultures from each strain were established in triplicate with an initial cell density of 500 cells mL^−1^, and growth was monitored every other day. The 2 mL samples were taken aseptically from each flask, fixed with acidic Lugol’s solution, and 1 mL was counted in a Sedgwick-Rafter counting chamber under an optical microscope (Primo Star, Zeiss, Jena, Germany) at 100× magnification.

Dinoflagellate biomass was harvested in the late exponential phase and collected in 50 mL centrifuge tubes. The tubes were centrifuged at 4000× *g* for 10 min at 4 °C, and the supernatant was discarded. The cell pellets were frozen at −80 °C, then lyophilized (Freezone 4.5 plus, Labconco, Kansas City, MO, USA), and kept frozen at −20 °C until extraction.

### 5.2. Methanolic Extraction of A. eilatiense (AeSQ181) Biomass

Dry cell pellets were weighed and then ground manually in a mortar with a pestle for 10 min. The biomass was sonicated (QSONICA, CT, USA) with 200 mL of methanol (MeOH, HPLC grade) in an ice bath for 1 min for five cycles and left to macerate for 120 h. This sonication step was repeated, and the extract was then centrifuged at 11,000× *g* for 45 min at 4 °C and filtered through 0.22 µm pore-size nylon membranes (DS0215-4020, Thermo Scientific, Waltham, MA, USA). The extract was evaporated at 40 °C and 337 mbar in a rotavapor (Buchi R-114, Flawil, Switzerland). For complete drying, the sample was lyophilized; for freezing, tri-distilled water MeOH:H_2_O (7:3 *v*/*v*) was added, then the sample extract was frozen at −80 °C and lyophilized. The dried methanol-free extract was kept at −20 °C until the bioassays were performed.

### 5.3. Fractionation of A. eilatiense (AeSQ181) Methanolic Extract

The methanolic extract of *A. eilatiense* AeSQ181, the strain that showed the highest cytotoxicity on MDA-MB-231 and H1563 cell lines, was partitioned and fractionated. The freeze-dried methanolic extract (3.0 g) was subjected to a liquid–liquid partition with 100 mL of ethyl acetate and 100 mL of water (×3) (EtOAc:H_2_O 1:1 *v*/*v*). The non-polar (EtOAc) and polar (H_2_O) phases were rotary-evaporated at 40 °C and weighed once dried.

Subsequent fractionation of the non-polar phase was performed by Flash Sephacore Chromatography (Pure C810 Flash, Buchi, Flawil, Switzerland). The stationary phase was silica gel 60 (230–400 mesh, Sigma-Aldrich, St. Louis, MO, USA). The mobile phase was hexane:EtOAc with a linear gradient ranging from 20 to 100% EtOAc over 42.5 min. The chromatography conditions included a run time of 75 min at a flow rate of 30 mL min^−1^. Detection was performed at a wavelength (λ) 240 nm to 800 nm, with absorbance measured across this range (Buchi, Flawil, Switzerland). The fractions were collected in 40 mL glass tubes.

The polar phase was fractionated in an open glass column (62 × 2 cm) packed with Sephadex (LH-20, Pharmacia Biotech, Uppsala, Sweden). The mobile phase was MeOH (ACS grade, Merck, Darmstadt, Germany); the fractions were collected in 10 mL glass tubes. Collected tubes from both polar and non-polar phases were examined by thin-layer chromatography (TLC) to identify and put together the fractions. Separations were performed on silica gel 60 F254 with fluorescent indicator plates (KGaA, Merck, Darmstadt, Germany). The plates were visualized at 254 and 365 nm wavelengths and developed with ceric sulfate. The fractions were placed in a fume hood to evaporate until solvent-free and kept at room temperature (23–25 °C) until chemical analyses were performed.

After the second bioassay series performed on MDA-MB-231 and H1563 cells, the fractions showing the highest cytotoxic activity (non-polar fractions FNP-5 and FNP-6) were selected for a second fractionation. Fractions were re-fractionated in an open glass column (62 × 2 cm) packed with Sephadex (LH-20, Pharmacia Biotech, Uppsala, Sweden). The mobile phase was MeOH (ACS grade, Merck, Darmstadt, Germany), and subfractions were collected in 10 mL glass tubes, which were separated by TLC on silica gel as previously reported. The samples were placed in a fume hood to evaporate until solvent-free and kept at room temperature (23–25 °C) until further chemical analyses were performed.

### 5.4. Cytotoxicity Assays

#### 5.4.1. Cell Culture and Conditions

Cytotoxicity assays were performed against breast and lung cancer cell lines purchased from the American Type Culture Collection (ATCC, Manassas, VA, USA). These cell lines comprised: human breast adenocarcinoma MCF-7 ATCC HTB-22 and MDA-MB-231 ATCC CRM-HTB-26; human breast carcinoma T47D ATCC HTB-133; human lung carcinoma A549 ATCC CRM-CCL-185 and H661 ATCC HTB-183; and human lung adenocarcinoma H1437 ATCC CRL-5872 and H1563 ATCC CRL-5875), with human fibroblast normal cells MRC-5 ATCC CCL-171 serving as controls.

Cell monolayers were cultivated in cell culture flasks in RPMI 1640 medium (Corning, New York, NY, USA) for H661, H1437, H156, and T47D; in MEM Eagle medium (Gibco, Gran Island, NY, USA) for MCF-7 and MRC-5; and DMEM medium (Cytiva, South Logan, UT, USA) for MDA-MB-231 and A549. All media were supplemented with 10% Fetal Bovine Serum (FSB) (Biowest, Nuaillé, France) and 1% antimycotic-antibiotic (streptomycin sulfate and sodium Penicillin G, Gibco, Waltham, MA, USA).

#### 5.4.2. Cell Viability Assays

Cell viability was assessed using a colorimetric MTS assay (3-(4,5-dimethylthiazol-2-yl)-5-(3-carboxymethoxyphenyl)-2-(4-sulfophenyl)-2H-tetrazolium), which was divided into three phases. The first phase involved evaluation of the methanolic extracts in the seven cell lines; the second and third phases evaluated the non-polar and polar fractions using the most sensitive cell lines to the previous extracts.

Cell cultures were incubated at 37 °C in a humidified atmosphere with 5% CO_2_ until they reached 80% confluence. Cells were detached using 1 mL of 0.25% Trypsin-EDTA (Gibco, Grand Island, NY, USA) for 1 min, followed by incubation at 37 °C without CO_2_. The cells were then seeded at a density of 5 × 10^4^ cells per well in 96-well flat-bottom plates and incubated at 37 °C for 24 h. After this period, the medium was removed and replaced with medium containing the *A. eilatiense* extract (methanol-free), the solvent-free fraction, or subfraction (all dissolved in 0.1% DMSO) (Sigma-Aldrich, St. Louis, MO, USA). The final concentration was 100 µg mL^−1^ for the methanolic extract and 50 µg mL^−1^ for the polar or non-polar fractions (FP and FNP) and the non-polar subfractions.

DMSO served as the administration vehicle for all bioassays in our experimental design because it dissolves both polar and non-polar compounds due to its amphipathic nature and exhibits no biological activity at low concentrations. However, at high concentrations (>5%), DMSO significantly decreases cell viability due to its cytotoxicity, leading to cell death [65,66]. DMSO is often used as an administration vehicle for cell penetration and as a proxy with crude extracts to allow for comparable results, with complex mixtures of unidentified compounds with unknown biological activities.

Regarding the subfractions, paclitaxel served as a positive control since this study aimed to identify cytotoxicity effects on cancer cells. Given that subfractions are less complex mixtures, they are anticipated to produce effects more akin to those of a specific antineoplastic. Thus, a compound with a more specific response was used as the control. To evaluate cell viability in the third phase tests with subfractions FNP51-53 and FNP61-62, 0.1% DMSO acted as the negative control, while paclitaxel from *Taxus brevifolia,* with a purity of ≥95% (Sigma-Aldrich, CAS 33069-62-4, St. Louis, MO, USA), at a concentration of 500 nmol L^−1^, was utilized as the positive control. Untreated cells were included to compare the effect (if any) of 0.1% DMSO on the cells.

Plates were incubated for 48 h. Once the treatment time elapsed, 10 µL MTS solution were added to each well, and the plate was further incubated for 3 h. Absorbance was read at 492 nm with a microplate reader (EPOCH Bioteck, Winooski, VT, USA). Cell viability was calculated as a function of the changes induced by the cytotoxicity in the treatments, and it was expressed as a percentage of the healthy cells compared to the negative control.

The selectivity index (SI) is a quantitative measure to assess the differential cytotoxicity of a compound between normal and cancerous cell lines. Herein, the quotient of normal cell viability percentage was calculated by dividing it by the cancerous cell viability percentage. An SI value exceeding 1 signifies that the compound exhibits a greater cytotoxic effect on the tested cancer cell line compared to its normal cellular counterpart. This metric is crucial in evaluating the therapeutic potential of compounds, as it indicates a preference for targeting malignant cells over healthy ones [67].

Three independent experiments tested each extract or fraction in triplicate. Statistical analyses were performed in GraphPad Prism 5 (GraphPad Software, San Diego, CA, USA). Data were analyzed by one-way ANOVA followed by Tukey’s HSD test; *p*-values ≤ 0.05 were considered statistically significant.

### 5.5. Liquid Chromatography Coupled to High-Resolution Mass Spectrometry Analysis (LC-HRMS)

The freeze-dried non-polar subfractions FNP-5 and FNP-6 were resuspended in MeOH (LC-MS grade) for preliminary LC-HRMS analysis. Chromatographic separation was performed with a HSS T3 column (Waters, Milford, MA, USA) (100 × 2.1 mm; 1.7 µm) maintained at 40 °C, with the mobile Phase A (100% H_2_O) and mobile phase B (100% acetonitrile), both containing 0.1% formic acid. The starting conditions were 100% A held for 4 min and increased to 98% B at 21 min, with a cycle time of 30 min. The injection volume was 2 µL. The effluent was directed into the HESI probe from 0.2 to 25 min. HRMS screening was performed with Orbitrap Exploris 120 mass spectrometer coupled to a Vanquish UPLC system (ThermoFisher, Hemel Hempstead, UK). Data were acquired separately in positive and negative ionization modes at a resolution of 120,000 scanning from either 75–1000 or 150–1500 amu with a capillary voltage of either +3500 or −2500 V. Sheath gas was set to 50, Aux Gas 10, and ion transfer tube and vaporizer temperatures of 325 and 350 °C, respectively. For data-dependent MS2 experiments, the four most intense ions selected from a full scan at a 60,000 resolution were subjected to fragmentation using normalized collision energies of 20, 40, and 100% and analyzed at a resolution of 15,000. For all experiments, an internal calibrant was used at the start of each run with a lock mass of *m*/*z* 203.0855.

### 5.6. Untargeted Metabolomic Analysis

The datasets were analyzed with a methodology based on Hernández-Melgar et al. 2024 [68] using open-access software and online tools. The main steps were: 1. MZmine 3.9.0 software was applied for feature extraction and alignment; and 2. metabolite identification was performed by matching the acquired MS2 spectra against reference spectra with the Global Natural Products Social Molecular Networking (GNPS) web platform. For those metabolites without automatic spectral matches, in silico annotations were proposed using MolDiscovery version 1.0.0 [69], CSIFinger:ID version 6.0.6 [70], DEREPLICATOR+ version 1.0.0 [71], and MS-FINDER version 3.6.1 [72,73]. These enabled the structural classification of metabolites at Levels 2 and 3, as defined by the Metabolomics Standards Initiative [74]. Chemical class assignments were made using the CANOPUS tool [75] integrated with CSIFinger:ID (SIRIUS) software version 5.8.5 [76] using the NPClassifier system.

Feature-based molecular networking (FBMN), in conjunction with the Metabolomics Standards Initiative (MSI Levels 2 and 3), were also applied. For positive mode annotations, GNPS and computational tools such as CSIFinger:ID, DEREPLICATOR+, MolDiscovery and Comprehensive Marine Natural Products Database (CMNPD) were used. For negative mode annotations, GNPS and computational tools like CSIFinger:ID (SIRIUS) and MSFinder were employed. In both ionization modes, the generated molecular networks grouped structurally related metabolites that share chemical characteristics into chemical families (nodes) based on their similar scores (cosine scores > 0.7).

For the construction of molecular networks for fractions FNP-5 and FNP-6, samples were aligned on each ionization mode, the spectral library, and the abundance matrix of features (https://drive.google.com/drive/folders/15ZIzuliwD_dl8DHlNR26XXSgmrURusyC?usp=drive_link, accessed on 5 January 2025). These spectral libraries, containing the automatic annotation from the Global Natural Products Social Molecular Networking (GNPS) (MSI, Level 2), were enriched with the in silico annotations (MSI, Level 3).

For additional details on the processing parameters for each approach, consult the Appendix A.

## Figures and Tables

**Figure 1 toxins-17-00150-f001:**
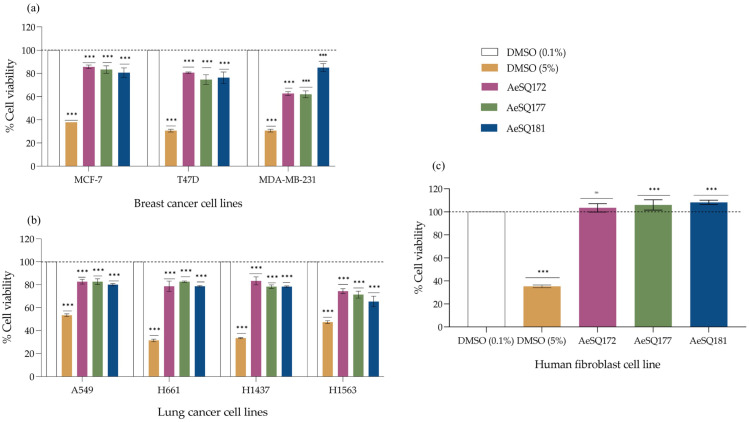
The cytotoxic effect of solvent-free crude methanolic extracts of the three *A. eilatiense* strains on (**a**) breast cancer cell lines; (**b**) lung cancer cell lines; and (**c**) fibroblast normal cell lines. The negative control and administration vehicle was 0.1% DMSO; the positive control corresponds to 5% DMSO. Exposure time 48 h. Values are represented as mean ± SD, *n* = 3 (*** *p*-value ≤ 0.001, one-way ANOVA, Tukey’s HSD test). ns: no significant difference. Note: The positive control in MCF-7 does not have error bars due to the absence of variability (standard deviation = 0). The dotted line marks 100% of viability in control group.

**Figure 2 toxins-17-00150-f002:**
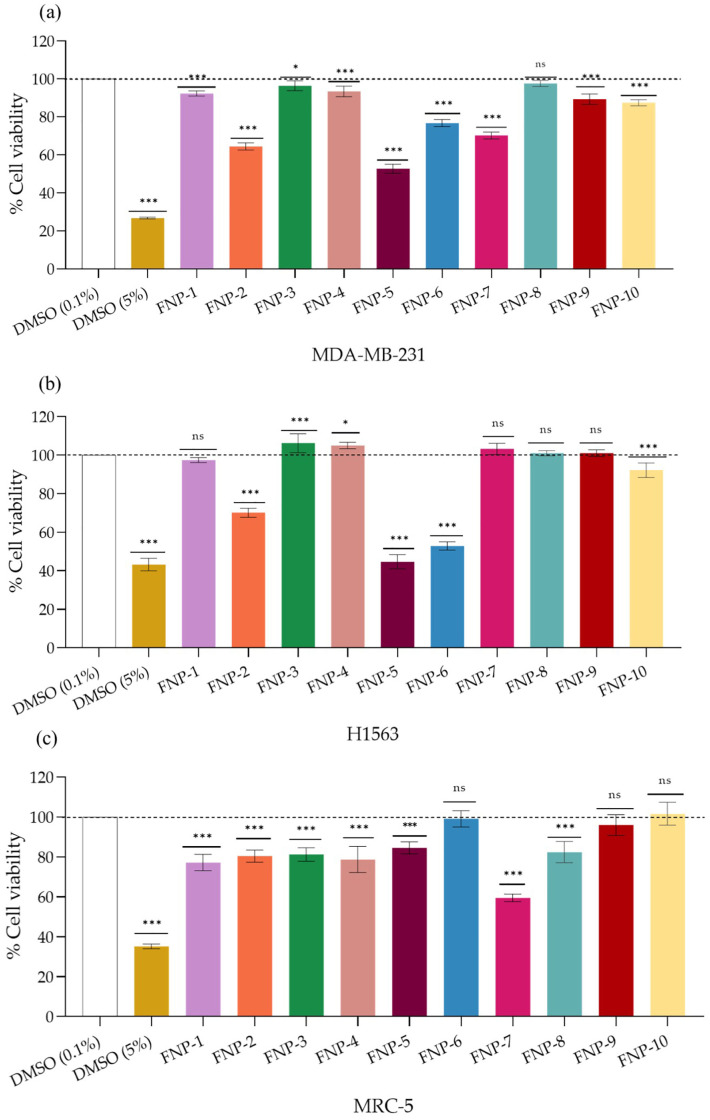
Cytotoxic effect of the non-polar fractions (FNP) of *A. eilatiense* AeSQ181 on the cancer cell lines. (**a**) Breast cancer MDA-MB-231; (**b**) lung cancer H1563; and (**c**) fibroblasts normal cells MRC-5. Treatment concentration: 50 µg mL^−1^. The negative control and administration vehicle were 0.1% DMSO. Positive control was 5% DMSO. Exposure time 48 h. Values are represented as mean ± SD; *n* = 3. * *p*-value ≤ 0.05,*** *p*-value ≤ 0.001, and ns: no significant difference compared to the control. One way ANOVA followed by Tukey’s HSD test. The dotted line marks 100% of viability in the control group.

**Figure 3 toxins-17-00150-f003:**
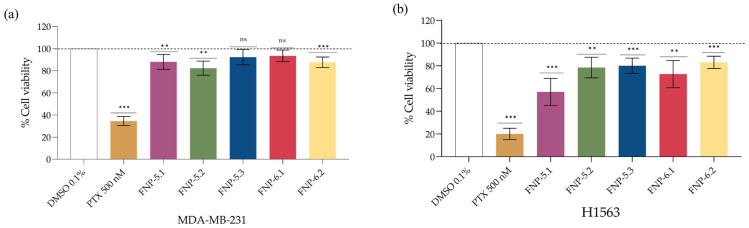
Effect of FNP-5 and FNP-6 sub-fractions on (**a**) breast cancer MDA-MB-231 and (**b**) lung cancer H1563 cell viability with treatments applied at 50 µg mL^−1^. The negative control and administration vehicle was 0.1% DMSO. Positive control corresponds to cells treated with paclitaxel (PTX) at 500 nmol L^−1^. Exposure time 48 h. Values are represented as mean ± SD; *n* = 3. One way ANOVA followed by Tukey’s HSD test. ** *p*-value ≤ 0.01, *** *p*-value ≤ 0.001, and ns: no significant difference. The dotted line marks 100% cell viability in the control group.

**Figure 4 toxins-17-00150-f004:**
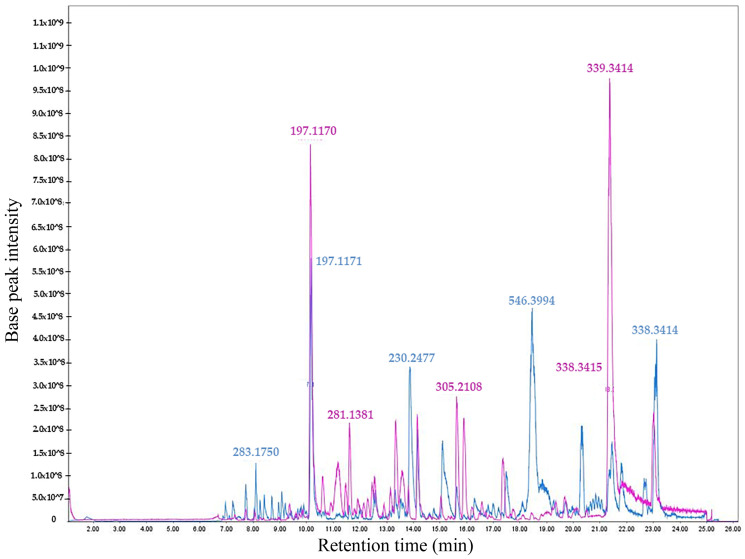
Total Ion Chromatograms (TIC) showing the overlap of the FNP-5 (pink) and FNP-6 (blue) fractions in positive ionization mode. The most intense signals are highlighted with their corresponding *m*/*z* ratios.

**Figure 5 toxins-17-00150-f005:**
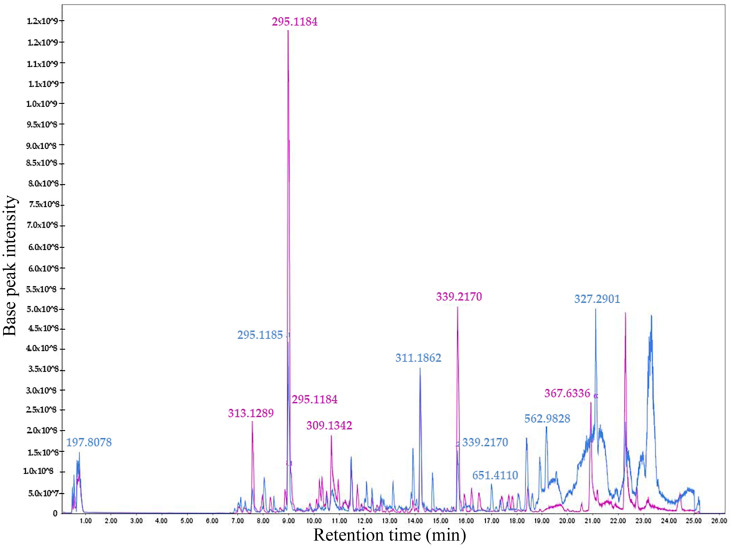
The total ion chromatograms (TIC) showing the overlap of FNP-5 (pink) and FNP-6 (blue) fractions in the negative ionization mode. The most intense signals are indicated with their corresponding *m*/*z* ratios.

**Figure 6 toxins-17-00150-f006:**
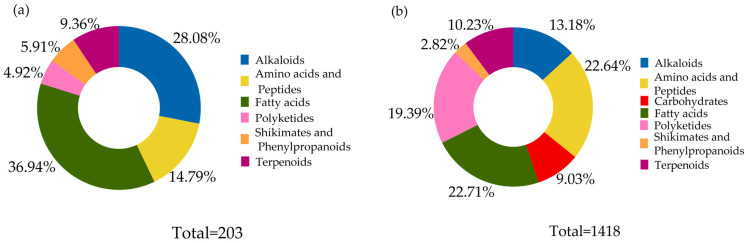
Distribution of metabolic pathways of chemical compounds detected in positive ionization mode for the bioactive fractions: (**a**) FNP-5 and (**b**) FNP-6 of *A. eilatiense* AeSQ181.

**Figure 7 toxins-17-00150-f007:**
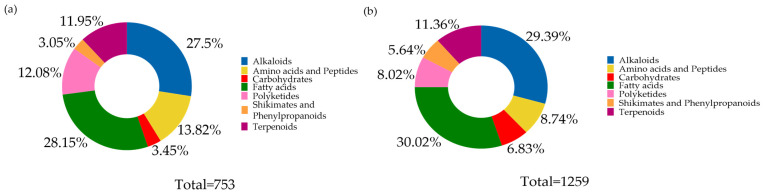
Distribution of metabolic pathways of chemical compounds detected in negative ionization mode for the bioactive fractions (**a**) FNP-5 and (**b**) FNP-6 of *A. eilatiense* AeSQ181.

**Figure 8 toxins-17-00150-f008:**
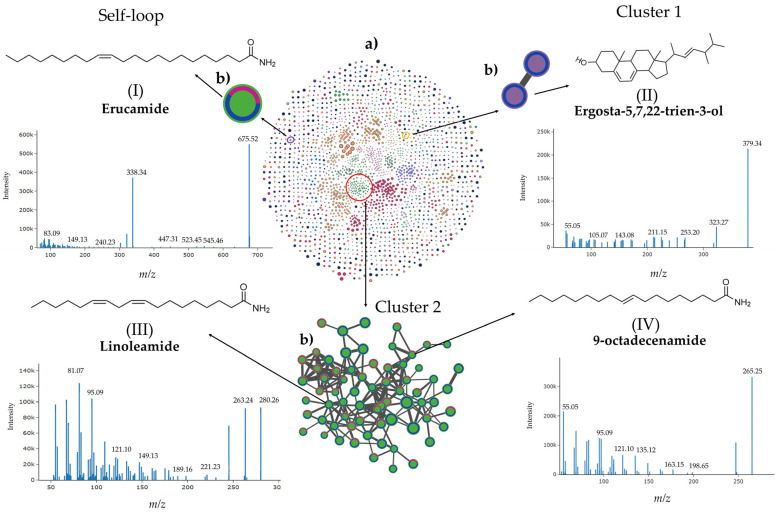
Molecular network of the bioactive subfractions FNP-5 and FNP-6 from *A. eilatiense* AeSQ181 strain obtained by LC-HRMS in positive ionization mode (**a**) (center). Clusters (sub-networks) and self-loop, both containing compounds with cytotoxic activity against cancer cell lines (**b**). The abundance of the features present in each active fraction is represented by color: pink for FNP-5 and blue for FNP-6. Roman numerals correspond to the selected compounds. Fragmentation mass spectra from tandem mass spectrometry (MS/MS) and assigned chemical structures of each compound of interest are also shown. The lines represent the cosine score values. The thicker the line, the higher its cosine score value; therefore, the higher the spectral similarity among connected features.

**Figure 9 toxins-17-00150-f009:**
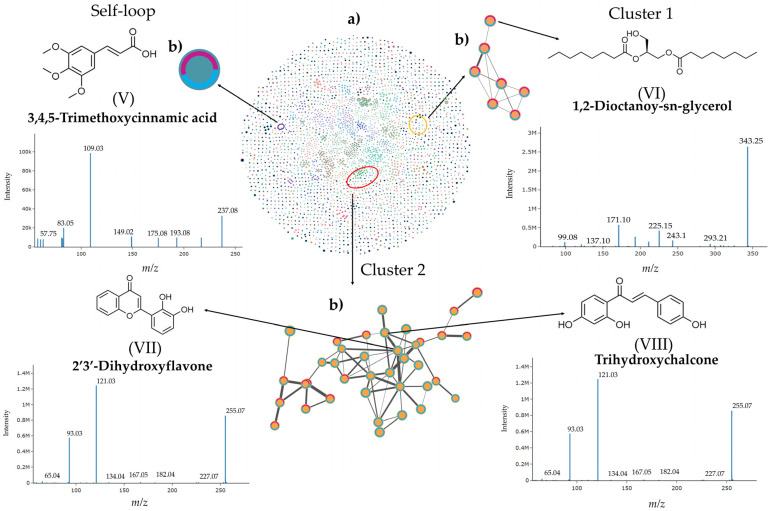
Molecular network of the bioactive subfractions FNP-5 and FNP-6 from *A. eilatiense* AeSQ181 obtained by LC-HRMS in negative ionization mode (**a**) (center). Clusters (sub-networks) and self-loop, both containing compounds with cytotoxic activity against cancer cell lines (**b**). The abundance of the features present in each active fraction is represented by colors: pink for FNP-5 and blue for FNP-6. Roman numerals correspond to the selected compounds. Fragmentation mass spectra from tandem mass spectrometry (MS/MS) and assigned chemical structures of each compound of interest are also provided. The lines represent the cosine score values. The thicker the line, the higher its cosine score value; therefore, the higher the spectral similarity among connected features.

**Table 1 toxins-17-00150-t001:** Selectivity index (SI) values of extracts for breast and lung cancer cell lines. The highest values are in bold.

Cell Line		Methanolic Extract		
		AeSQ172	AeSQ177	AeSQ181
Breast	MCF-7	1.20	1.28	**1.35**
	T47D	1.27	1.41	**1.42**
	MDA-MB-231	1.63	**1.71**	1.27
Lung	A549	1.24	1.28	**1.35**
	H661	1.30	1.28	**1.37**
	H1437	1.32	**1.36**	1.30
	H1563	1.39	1.49	**1.66**

**Table 2 toxins-17-00150-t002:** Selectivity index (SI) values of extracts for breast (MDA-MB-231) and lung (H1563) cancer cell lines. The highest values are in bold.

Fractions	SI MDA-MB-231	SI H1563
FNP-1	0.84	0.79
FNP-2	1.25	1.14
FNP-3	0.84	0.81
FNP-4	0.85	0.75
**FNP-5**	**1.63**	**1.93**
**FNP-6**	**1.29**	**1.87**
FNP-7	0.84	0.57
FNP-8	0.84	0.81
FNP-9	1.08	0.95
FNP-10	1.16	1.1

## Data Availability

The raw LC-MS2 data sets were deposited in the GNPS/Massive public repository under accession number MSV000097363, accessed on 18 March 2025. Feature based-molecular networking in positive ionization mode (https://gnps.ucsd.edu/ProteoSAFe/status.jsp?task=b17a830d74294bb399d2306e984c4c32, accessed on 2 December 2024). Feature based-molecular networking in negative ionization mode (https://gnps.ucsd.edu/ProteoSAFe/status.jsp?task=1d59a5a7c29a44e7a045ee25a3fe751d, accessed on 10 December 2024). METABOLOMICS-SNETS-V2 of FNP-5 (https://gnps.ucsd.edu/ProteoSAFe/status.jsp?task=56c5cccf9e194b5ca2588db6938b9441, accessed on 7 November 2024) of FNP-6 (https://gnps.ucsd.edu/ProteoSAFe/status.jsp?task=3b0747e820664e7fa1621a078d1e14bc, accessed on 7 November 2024). DEREPLICATIOR+ of FNP-5 and FNP-6 (https://gnps.ucsd.edu/ProteoSAFe/status.jsp?task=ad184dfa6e4d4bf4a88dbf2ff5de7ab3, accessed on 2 December 2024). MOLDISCOVERY of FNP-5 and FNP-6 (https://gnps.ucsd.edu/ProteoSAFe/status.jsp?task=cb79def02dcb4729923a89648a4a39d4, accessed on 2 December 2024).

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
