# Peer review of "Untargeted Metabolomic Analysis and Cytotoxicity of Extracts of the Marine Dinoflagellate Amphidinium eilatiense Against Human Cancer Cell Lines"

_toxins, 2025, doi:10.3390/toxins17040150_

Round 1

Reviewer 1 Report

Comments and Suggestions for Authors

This manuscript investigated the cytotoxicity of dinoflagellate Amphidinium eilatiense to human cancer cells lines , including 3 human breast cancer cell lines and 4 lung cancer cells lines. And the cell viability of these 7 cell lines were detected after 48 h exposure to the crude methanolic extracts from 3 strains. This work further analyzed the cytotoxic franctions by liquid chromatography coupled with high-resolution mass spectrometry. The results revealed these strains included a much richer chemical diversity profile. These results are acceptable. However, my biggest reservation is the method to test cytotoxicity. The cell viabilities of 7 cell lines were selected to indicate dinoflagellate-induced toxicity. It is a preliminary result. I think it would be better for more parameters to be checked to indicate cytotoxicity. I strongly suggest that the authors consider testing more parameters to reveal the cytotoxicity of extracts of this species. I would like to re-review this manuscript after appending some work.

Author Response

Comments and Suggestions for Authors

This manuscript investigated the cytotoxicity of dinoflagellate Amphidinium eilatiense to human cancer cells lines, including 3 human breast cancer cell lines and 4 lung cancer cells lines. And the cell viability of these 7 cell lines were detected after 48 h exposure to the crude methanolic extracts from 3 strains. This work further analyzed the cytotoxic franctions by liquid chromatography coupled with high-resolution mass spectrometry. The results revealed these strains included a much richer chemical diversity profile. These results are acceptable.

Authors’ reply: The authors value the reviewer’s feedback.

However, my biggest reservation is the method to test cytotoxicity. The cell viabilities of 7 cell lines were selected to indicate dinoflagellate-induced toxicity. It is a preliminary result. I think it would be better for more parameters to be checked to indicate cytotoxicity. I strongly suggest that the authors consider testing more parameters to reveal the cytotoxicity of extracts of this species. I would like to re-review this manuscript after appending some work.

Authors’ reply: We greatly appreciate the reviewer’s comments and agree that conducting additional tests to evaluate cytotoxicity would be very valuable. But the reviewer is non-specific about what other toxicity tests should be required. In any case, the experimental design was to establish the basis for a functional screening assay for bioassay-guided fractionation with various plausible cancer cell lines. The data provide the relevant basic information.

We calculated the selectivity index (SI) values based on the cytotoxicity results, allowing us to add two new tables (Tables 1-2). Additionally, we included a detailed explanation in the "Results" section (lines 158-166 and 238-240) to clarify how these values complement the data obtained thus far. Here, we respond in detail to the reviewer’s concern about the limitations of the toxicological tests and how we have followed a stepwise interpretation to adapt our experimental design: based on the selectivity index (SI) results, the AeSQ181 methanolic extract was optimal for bioassay-guided fractionation for the following reasons: 1) AeSQ181 is the highest biomass producer among all three strains. This is important because sample size may decrease after each step in the fractionation, which affects yield; 2) as indicated by the selectivity index results, the AeSQ181 strain exhibits higher selectivity against 5 out of 7 cancer cell lines tested (Tables 1 and 2). This finding supports the evidence observed in cytological tests; 3) of the three strains tested, AeSQ181 demonstrated the least cell damage in biological tests on normal cells (MRC-5). This is a crucial factor to consider when assessing natural products for further testing and drug development.

Finally, fractions FNP-5 and FNP-6 exhibited the highest SI values for the MDA-MB-231 cell line (breast) and the H1563 cell line (lung), with some values nearly reaching an index of 2 (Table 2). Consequently, both fractions were selected for the sub-fractioning stage – we adhered to the classic stepwise selection for “bioassay-guided fractionation." In our view, the results presented in this paper demonstrate that the objective of optimizing the screening assay was accomplished.

Further cytotoxicological experiments are currently not feasible due to insufficient new biological material, which can take months to years to generate for additional assays. Additionally, we cannot immediately access the original conditioned cell lines. Nevertheless, we propose conducting further cytotoxicological testing in upcoming studies when we obtain more HR-MS confirmation of the metabolomic components from these fractionated extracts of Amphidinium eilatiense, including the potential role of amphidinols.

Reviewer 2 Report

Comments and Suggestions for Authors

Overall, this is a well written manuscript. I have one minor comment.

For figures 1-7, adding a dotted horizontal line across 100% cell viability will be helpful to show where the statistically significant data are decreasing since some of the data increase from the control. This particularly caught my attention in figure 3. There are asterisks above some of the bars, but the text notes no significant decrease. This makes sense when combining the text and figure, but if the figure were standing alone, one may mistake the significance as a decrease.

Author Response

Comments and Suggestions for Authors

Overall, this is a well written manuscript.

Authors’ response: The authors sincerely appreciate the reviewer’s favorable opinion on our manuscript.

I have one minor comment.

For figures 1-7, adding a dotted horizontal line across 100% cell viability will be helpful to show where the statistically significant data are decreasing since some of the data increase from the control. This particularly caught my attention in figure 3. There are asterisks above some of the bars, but the text notes no significant decrease. This makes sense when combining the text and figure, but if the figure were standing alone, one may mistake the significance as a decrease.

Authors’ response: We greatly appreciate the reviewer’s keen observations regarding the graphs in the manuscript. As suggested, we have added a dashed line to all graphs to make the data visualization clearer. Additionally, we have rewritten the paragraph that caused confusion, aiming to improve clarity and ensure the information is more comprehensible (Lines 152-156 in the revised version).

Reviewer 3 Report

Comments and Suggestions for Authors

The study “Untargeted Metabolomic Analysis and Cytotoxicity of Extracts of the Marine Dinoflagellate Amphidinium eilatiense against Human Cancer Cell Lines” presents an interesting approach to bioassay-guided fractionation and metabolomic analysis of A. eilatiense. However, several issues need to be addressed before the manuscript can be considered for publication. Below are major concerns:

  1. The manuscript states that three breast and four lung cancer cell lines were used, but there is limited discussion on why these specific lines were selected. It is crucial to provide justification regarding their biological relevance to A. eilatiense metabolites. Were these cell lines chosen based on prior literature showing responsiveness to similar compounds? A more detailed rationale should be included in the Introduction.

  1. The study presents percent inhibition data but does not report IC50 values for the crude extracts or fractions. These values are critical for comparing cytotoxic potency. Please include IC50 calculations or justify their omission. Furthermore, in the content of selectivity towards cancer vs. normal cells, the manuscript claims that the extracts did not significantly affect normal human fibroblasts (MRC-5). However, was selectivity quantified? A selectivity index (SI = IC50 Normal / IC50 Cancer) should be provided. The authors should also consider discussing the potential mechanism of selectivity, given that fatty acid-derived metabolites often affect membrane integrity.

  1. While the study uses GNPS-based molecular networking, there is no clear discussion of how fragmentation spectra were verified. Were authentic standards used? If not, what level of confidence (Metabolomics Standards Initiative) can be assigned to the compound identifications? The presence of unique metabolomic features in FNP-5 and FNP-6 is interesting, but their specific contributions to cytotoxicity are unclear. Please provide a more direct correlation between metabolomic features and bioactivity.

  1. The metabolomic analysis suggests the presence of known cytotoxic compounds (e.g., amphidinols, fatty acid amides), but their biological activity is inferred rather than directly tested. Were any of these compounds isolated and tested for cytotoxicity in pure form?

  1. From Fig. 1 to Fig. 7, it is difficult for me to understand the negative control (0.1% DMSO) and positive control (5% DMSO). Please state (in detail) why they are used as controls, rather than a standard chemotherapeutic (e.g., paclitaxel, doxorubicin). In addition, from Fig. 1 to Fig. 7, why the negative control (0.1% DMSO) has not error bar? Please explain.

  1. Since there are obvious homogenizations among 13 figures in this manuscript, the 13 figures are strongly suggested to be merged into six figures: Fig. 1 (1-3), Fig. 2 (4-5), Fig. 3 (6-7), Fig. 4 (8-9), Fig. 5 (10-11), Fig. 6 (12-13).

  1. The statistical analyses are not well explained. Please specify the type of ANOVA used (one-way or two-way) and provide effect sizes where applicable.
Comments on the Quality of English Language

 The English could be improved to more clearly express the research.

Author Response

Comments and Suggestions for Authors

The study “Untargeted Metabolomic Analysis and Cytotoxicity of Extracts of the Marine Dinoflagellate Amphidinium eilatiense against Human Cancer Cell Lines” presents an interesting approach to bioassay-guided fractionation and metabolomic analysis of A. eilatiense. However, several issues need to be addressed before the manuscript can be considered for publication. Below are major concerns:

  1. The manuscript states that three breast and four lung cancer cell lines were used, but there is limited discussion on why these specific lines were selected. It is crucial to provide justification regarding their biological relevance to  eilatiensemetabolites. Were these cell lines chosen based on prior literature showing responsiveness to similar compounds? A more detailed rationale should be included in the Introduction.

Authors’ response: We greatly appreciate your comments and annotations regarding selecting the cell lines used in this study. In the Introduction, we have expanded the justification for our choice of three breast and four lung cancer cell lines to address your observation (lines 85-87 in the revised version with track changes).

  1. The study presents percent inhibition data but does not report IC50 values for the crude extracts or fractions. These values are critical for comparing cytotoxic potency. Please include IC50 calculations or justify their omission. Furthermore, in the content of selectivity towards cancer vs. normal cells, the manuscript claims that the extracts did not significantly affect normal human fibroblasts (MRC-5). However, was selectivity quantified? A selectivity index (SI = IC50 Normal / IC50 Cancer) should be provided. The authors should also consider discussing the potential mechanism of selectivity, given that fatty acid-derived metabolites often affect membrane integrity.

Authors' response: Regarding the absence of IC50 values for the crude extracts and fractions, we regret that we were unable to obtain these values due to the limited biological material (dinoflagellate cell biomass) produced in culture, which restricted the number of experiments we could conduct. The experimental design involved starting with an initial concentration of the crude extracts and then adjusting based on the observed cytotoxic activity through bioassay-guided fractionation, leading to semi-purified fractions. However, we have determined and now provide the selectivity index values for the extracts and fractions, which are presented in two new tables (Tables 1 and 2, lines 182 and 243), along with a description of the methodology (lines 706-712).

Additionally, we have considered your comment regarding the biological activity of the lipids and have expanded the discussion on this aspect in lines 518-533. Here, we explore the potential mechanism of selectivity, especially concerning membrane integrity, since fatty acid-derived metabolites may affect the results obtained from the investigation:

  1. While the study uses GNPS-based molecular networking, there is no clear discussion of how fragmentation spectra were verified. Were authentic standards used? If not, what level of confidence (Metabolomics Standards Initiative) can be assigned to the compound identifications? The presence of unique metabolomic features in FNP-5 and FNP-6 is interesting, but their specific contributions to cytotoxicity are unclear. Please provide a more direct correlation between metabolomic features and bioactivity.

Authors’ response: Thank you for your valuable feedback. We will address each point individually.

Regarding the selection of the cancer cell lines, they were chosen because:

1) Previous studies indicate that extracts, fractions, and compounds of Amphidinium carterae exhibit cytotoxic activity against the MCF-7 and A549 cell lines. This dinoflagellate is closely related to Amphidinium eilatiense, raising the question of whether it might have a similar effect on these cancer cell lines. To address your concern, this information and the references have been added (lines 85-87).

2) The remaining cancer cell lines tested were selected to investigate similar effects across a wider variety of cell lines within the same type of cancer, as no previous studies exist involving such a diversity of cancer cell-types with various Amphidinium species/strains.

3) Regarding the verification of the fragmentation spectra, we have already included a discussion in lines 533-553. Additionally, we have assigned a confidence level in accordance with the Metabolomics Standards Initiative (MSI) (lines 747-760), which enables us to establish identifications with a greater degree of certainty, albeit with some limitations due to the nature of the study.

For the construction of molecular networks for fractions FNP-5 and FNP-6, the samples were aligned with each ionization mode, and the spectral library, along with the abundance matrix of features (Supplementary Material), was obtained. These spectral libraries, which included automatic annotations from Global Natural Products Social Molecular Networking (GNPS) (MSI, level 2), were enriched with the in silico annotations (MSI, level 3) described in the methodology.

Molecular networking is based on paired spectral alignment and utilizes a similarity algorithm for spectral cosine. The process involves searching for modified spectral similarities by comparing fragmentation spectra (MS2) of ions with identical m/z values and MS2 spectra adjusted by the same m/z subtraction result as the precursor ion.

While unique metabolomic features in FNP-5 and FNP-6 are intriguing, their specific contribution to cytotoxicity remains unclear. Additional information (lines 517-532) regarding the metabolomic profiles has been included. Still, it is important to note that we do not have a pure compound in this instance, making it challenging to establish a direct and definitive correlation between these metabolomic features and bioactivity. Previously, the discussion included a description of the biological activity of the detected metabolites, but further research is needed to better understand this relationship.

4. The metabolomic analysis suggests the presence of known cytotoxic compounds (e.g., amphidinols, fatty acid amides), but their biological activity is inferred rather than directly tested. Were any of these compounds isolated and tested for cytotoxicity in pure form?

Authors’ response: Isolation and testing of pure compounds was outside the scope of the experimental design. Unfortunately, such pure compounds could not be isolated to conduct direct cytotoxicity tests in these experiments, nor are they available from other external sources at this stage. Testing pure compounds is for later stage experimentation when the metabolomic profile is better defined.

  1. From Fig. 1 to Fig. 7, it is difficult for me to understand the negative control (0.1% DMSO) and positive control (5% DMSO). Please state (in detail) why they are used as controls, rather than a standard chemotherapeutic (e.g., paclitaxel, doxorubicin). In addition, from Fig. 1 to Fig. 7, why the negative control (0.1% DMSO) has not error bar? Please explain.

Authors’ response: We greatly appreciate your observations regarding the experimental design of the biological tests. We have expanded the information in the methodology section, specifically in lines 684-699. Additionally, in the graph where the data is presented, we have added a clarifying note regarding the error bars, indicating that the absence of error bars is due to zero variation (lines 167-168).

6. Since there are obvious homogenizations among 13 figures in this manuscript, the 13 figures are strongly suggested to be merged into six figures: Fig. 1 (1-3), Fig. 2 (4-5), Fig. 3 (6-7), Fig. 4 (8-9), Fig. 5 (10-11), Fig. 6 (12-13).

Authors’ response: Thank you for your valuable observation. We have made the necessary changes and combined the figures 1-7 into three figures (Figures 1-3). Due to the complexity of Figures 4, 5, 6, and 7 (with the new numbering) and the sizes of Figures 8 and 9, it was not feasible to merge these.  

7. The statistical analyses are not well explained. Please specify the type of ANOVA used (one-way or two-way) and provide effect sizes where applicable.

Authors’ response: The necessary changes have been made to the figure legends, including the addition of the appropriate legend:  One way ANOVA followed by Tukey’s HSD test.

The English language has been revised thoroughly by a native speaker.

Round 2

Reviewer 1 Report

Comments and Suggestions for Authors

Thanks for the author's responses. Yes, I can understand the limited samples to test the cytotoxicity. The response sounds good. And some more supplementary data were offered.

Reviewer 3 Report

Comments and Suggestions for Authors

I am happy to recieve these sincere feedback. Since the authors have addressed all of my concerns, I have no other questions. Valuable discussion and good job!